# Metrology for Indoor Radon Measurements and Requirements for Different Types of Devices

**DOI:** 10.3390/s24020504

**Published:** 2024-01-13

**Authors:** Andrey Tsapalov, Konstantin Kovler

**Affiliations:** National Building Research Institute, Faculty of Civil and Environmental Engineering, Technion—Israel Institute of Technology, Haifa 3200003, Israel; andrei.ts@technion.ac.il

**Keywords:** indoor radon, measurement protocol, metrology, QA/QC, temporal and instrumental uncertainties, conformity assessment, ISO/IEC, rational criterion, short-term tests, long-term tests

## Abstract

Indoor radon measurements have been conducted in many countries worldwide for several decades. However, to date, there is a lack of a globally harmonized measurement standard. Furthermore, measurement protocols in the US (short-term tests for 2–7 days) and European Union countries (long-term tests for at least 2 months) differ significantly, and their metrological support is underdeveloped, as clear mathematical algorithms (criteria) and QA/QC procedures considering fundamental ISO/IEC concepts such as “measurement uncertainty” and “conformity assessment” are still absent. In this context, for many years, the authors have been advancing and refining the theory of metrological support for standardizing indoor radon measurements based on a rational criterion for conformity assessment within the ISO/IEC concepts. The rational criterion takes into account the main uncertainties arising from temporal variations in indoor radon and instrumental errors, enabling the utilization of both short- and long-term measurements while ensuring specified reliability in decision making (typically no less than 95%). The paper presents improved mathematical algorithms for determining both temporal and instrumental uncertainties. Additionally, within the framework of the rational criterion, unified metrological requirements are formulated for various methods and devices employed in indoor radon measurements.

## 1. Introduction

Radon is one of the most hazardous carcinogens, contributing the majority (about half) of the overall risk among natural and anthropogenic atomic radiation sources [1]. The primary source of radiation risk for the population is exposure at home and in workplaces, as people spend up to 90% of their time indoors [2], where radon levels are always higher than in outdoor air. Unlike other sources of radiation, such as cosmic or terrestrial (earth) radiation, the impact of radon can be regulated to reduce risk. Therefore, the World Health Organization (WHO) [3] and the International Commission on Radiological Protection (ICRP) [4] recommend, and the International Atomic Energy Agency (IAEA) [5] and the European Basic Safety Standards (EU-BSS) [6] require, where possible, the maximal restriction of the annual average radon level in buildings, which should not exceed 300 Bq/m^3^. The National Reference Levels (RLs) vary in European countries due to differences in regional radon levels and usually range from 100 to 300 Bq/m^3^ [7]. In the US, indoor radon concentration is regulated through an Action Level, set at 4 pCi/L (148 Bq/m^3^) [8,9], which is a concept different from the RL [10].

Previously, it was believed [11] that radon measurements in buildings located in Radon Priority Areas (RPAs), typically covering relatively small areas of a country, should be a priority because such selective control ensures more effective mitigation of radon risk. However, the extensive experience gained in Sweden [12] and the US [8,9], as well as recent modeling research conducted in Germany [13], indicate the necessity of performing measurements in all areas of the country, regardless of the regional mean radon level in buildings. Finally, the same approach is recommended in ICRP Publication 126 [4]. The reason is that, due to the high spatial variability in indoor radon concentrations, high levels can also occur in areas not labelled RPA. This means that any existing building is a potential risk object and should, therefore, be surveyed [8,9].

At the same time, there are millions of buildings in different countries [14], necessitating the creation of conditions for large-scale surveys and the effective identification of hazardous buildings (with mean radon concentrations above the RL), according to paragraph 3.46 in IAEA SSG-32 [15]. Such conditions should ensure not only a satisfactory accuracy in radon concentration measurements but also the high reliability of decisions regarding the compliance of the tested rooms (and the entire building) with the normative (reference) level. However, to date, a harmonious measurement protocol for reliable decision making based on a rational conformity assessment criterion, coordinated at the international level, is still lacking. Moreover, the fairly effective US approach to reducing radon risk [10], based on short-term tests (2–7 days) [8,9], radically differs from the European approach based on long-term tests (at least 2 months) [16].

What accounts for not just a lack of harmony, but such a radical difference between the US and European approaches to regulating indoor radon? In our view, two main reasons could explain this: (1) a misalignment in prioritizing both indoor radon regulation and research planning in Europe, and (2) the neglect of fundamental ISO/IEC concepts in the attempt to standardize indoor radon measurements.

Indeed, an analysis of the literature published over the past decades indicates that in Europe [17], the priority of surveys is detailing the location of Radon Priority Areas (RPAs), as well as assessing and refining collective risks, rather than identifying hazardous buildings and implementing indoor radon mitigation, in contrast to the US, where mitigation has been carried out in several million buildings [18]. Moreover, the extensive results from short-term indoor radon tests in each US state allow for high-resolution detailing of RPAs [19] and assessing collective risks without the need for long-term measurements. This approach is not only theoretically justified [17] but practically confirmed by the fact that there are only minor differences between the indoor radon map based on short-term tests and the map refined based on a complete set of geological data [20]. Thus, the detailing of RPAs is quite achieved through a large number of indoor radon measurements, regardless of the duration of the tests [17].

In this context, it is crucial to reiterate that the effective radon regulation industry in the US is primarily achieved through the public’s participation in large-scale (mass) short-term indoor radon measurements in all states, aimed at identifying hazardous buildings, rather than detailing RPAs [17]. However, instead of refining the indoor radon measurement protocol based on the US approach and developing its metrological support, European radon specialists, including the metrological community, pin high hopes on the possibility of identifying hazardous buildings without conducting indoor radon measurements [21] by predicting annual average indoor radon levels based on building and environmental characteristics using modern machine learning algorithms [21,22,23,24,25]. Indeed, there has been a significant increase in publications on this topic in recent years, although some of them are not related to solving the problem of identifying hazardous buildings [22]. At the same time, among the recently published relevant studies based on machine learning, it is reported that a satisfactory accuracy in predicting mean indoor radon is only achieved at the municipal level, while *“accurate prediction of point estimates such as the conditional mean of indoor radon is not yet possible, which is consistent with many other studies”* [21].

The international standard ISO 11665-8:2019 [26] is designed to standardize indoor radon measurements for the conformity assessment of a room (building) with a norm. However, as a result of a clear misalignment in priorities, neither the outcomes of research over the past decades, including the recently completed European metrological project ‘MetroRADON’ (Metrology for Radon Monitoring, metroradon.eu), nor the planned outputs of the current European project ‘RadoNorm’ (Managing Risk from Radon and NORM, radonorm.eu), can be used as quantitative criteria in this international standard to improve it. Moreover, despite the international status, ISO 11665-8:2019 expresses only the European approach in radon regulation and is nearly identical to its 2012 predecessor [16].

In contrast to the evident advantages of the US regulation compared to the European one, there persists a lingering skepticism among many specialists (regulators) in using the short-term measurements [17]. A significant drawback shared by both US and European measurement protocols, including ISO 11665-8, is the failure to account for a temporal uncertainty of indoor radon. The temporal uncertainty, a critical factor, usually significantly exceeds instrumental uncertainty (if the test duration is less than half a year) and is 2–3 times the coefficient of variation (COV) commonly used to estimate temporal variations in indoor radon [16,27]. Consequently, neither the US nor the European indoor radon measurement protocols can be deemed reliable.

This problem significantly complicates the development of a harmonized international standard. Despite the last version (2019) and recently initiated (2021) revisions of ISO 11665-8 [16], and the release of updated AARST/ANSI radon US standards in 2023 [28], this issue persists, notably overlooking fundamental ISO/IEC concepts such as ‘measurement uncertainty’ [29,30] and ‘conformity assessment’ [31]. It is apparent that, without the application of the rational concepts, the effective implementation of QA/QC into indoor radon measurements is unattainable. The application of fundamental ISO/IEC concepts, for example, in the more advanced US regulation [28], could substantially reduce the costs of metrological support for indoor radon measurements, ensuring better decision-making reliability. Moreover, it is important to highlight that the Action Level used in the US regulation is not analogous to the Reference Level, though these control levels share a certain connection [10].

Thus, the authors express concern about the observed misalignment in priorities within the radon scientific community in terms of the research into, and regulation of, indoor radon. Hence, the primary objective of this paper is to refocus the attention of the radon community on the actual needs for the development and implementation of harmonized international standardization of indoor radon measurements. This should be based on a rational criterion for conformity assessment of a room with a norm, ensuring a specified reliability in decision making by quantifying both temporal and instrumental uncertainties. A more detailed discussion of these two main components of the combined uncertainty of annual average radon concentration will provide a better understanding of the metrological needs for the effective implementation of QA/QC into indoor radon measurements.

## 2. The Rational Criterion for Conformity Assessment

For the first time, the rational criterion for conformity assessment was introduced at Final Symposium COST NETWORK “NORM4Building” in Rome (6–8 June 2017) over 6 years ago. The authors have been consistently refining, and striving to promote, the rational criterion through publishing the results of their studies in journals [10,16,17,27,32] and presentations at conferences, for example, (i) the 29th Conference of the Nuclear Societies in Israel (6–8 May 2018), (ii) the Ninth International Symposium on Naturally Occurring Radioactive Material (NORM IX, 23–27 September 2019, Denver, CO, USA), (iii) a EURAMET EMPIR meeting (PRT MetroRadon II, 22–23 January 2020, BEV, Vienna, Austria), (iv) European Radon Week 2020 (24–28 February 2020, BEV, Vienna, Austria), (v) the 2020 VIRTUAL International Radon Symposium, Plug in to Recharge (9–13 November 2020, AARST, US), (vi) the 6th European Congress on Radiation Protection (30 May–3 June 2022, Budapest, Hungary), (vii) AARST Symposium (24–26 October 2022, Bellevue, WA, USA), and (viii) RAD 11 (19–23 June 2023, Herceg Novi, Montenegro). In addition, the authors actively participate in the ISO working group [16]. The standardization of indoor radon measurements based on a rational criterion for conformity assessment has been supported and promoted by the leading radon experts in the Russian Federation [33]. However, despite our efforts, the rational criterion has not yet been recognized and is not being implemented in either European or US regulatory practice due to the deeply conservative beliefs of regulatory leaders. Unfortunately, instead of a harmonization of the measurement protocol and an increase in the reliability of decision making, the preference is given to highly ineffective (irrational) traditional approaches to indoor radon testing [10] that have been deeply entrenched over several decades [16,27].

The rational criterion for conformity assessment of a room with a norm at a given (manageable) reliability of decision making (at least 95%, no more than 5% false-negative error) for both short- and long-term measurements, which is suitable to improve both European and US practices, is expressed as follows [16,27]:(1)$$C(t)∙[1+\sqrt{{U_{V}(t)}^{2}+{U_{D}}^{2}}]<C_{RL}$$
where *C*(*t*) is the measured radon concentration over the test period of *t* (*t ≥* 2 days), *C_RL_* is the reference (normative) level for the annual average indoor radon concentration, *U_D_* is the instrumental (device) uncertainty (with a coverage factor of 2) that combines all sources of uncertainty (mainly random and systematic/calibration components) associated with the measured radon concentration, regardless of the nature of radon origin and the behavior of radon in time and space, *U_V_*(*t*) is the temporal uncertainty of indoor radon, defined as the value of the 95th percentile (or 95% probability) in the distribution of all deviations between the measured concentrations *C_ij_*(*t*) and the annual average concentration: *D_ij_*(*t*) = *C_j_^AA^/C_ij_*(*t*) − 1 (*i* = 1…*M*; *j* = 1…*N*), in a representative sample of *N* buildings (rooms) within an international or national case study. In each of *N* buildings (rooms), year-long continuous measurements (YLCMs) with a registration period of 1 or 3 h [27] (at *M* = 8760 or 2920, respectively) of the radon concentration are carried out and provide good statistics from the arrays *D_ij_*(*t*) for any measurement duration *t* [27].

Traditionally, the scientific community has primarily focused on the study and validation of instrumental uncertainty (*U_D_*), while the temporal (key) uncertainty *U_V_*(*t*) has only recently garnered attention, previously substituted by surrogate parameters such as the seasonal correction factor (SCF) and coefficient of variation (COV) [16]. Unfortunately, this long-standing substitution has led to a significant underestimation of the role of temporal variations in indoor radon within the current regulations [27]. In fact, *U_V_*(*t*) is considerably larger than *U_D_*, not only in the case of short-term but also long-term tests, as long as their duration does not exceed 6–8 months [10,16,27,32]. The *U_D_* parameter in criterion (1) can be considered better known, as its values comprise a mandatory characteristic of a specific device, which must be periodically checked by the National Metrology Institute. Therefore, the scientific community’s primary focus within the current metrological needs for indoor radon measurements should shift towards the *U_V_*(*t*) parameter instead of *U_D_*. The following sections provide a more detailed description of both temporal and instrumental uncertainties to better understand the metrological requirements for QA/QC in indoor radon measurements.

An additional source of uncertainty in conformity assessment may arise from year-to-year variations in indoor radon, driven by long-term climate variations, occupant behavior, and the degradation or restoration of building structures, including underground utilities. However, accumulated data over many years indicate that, in most cases, year-to-year variations in indoor radon are insignificant, typically not exceeding 15–20% [34], serving as a benchmark for the maximum achievable accuracy within the framework of criterion (1). Nevertheless, it is beneficial to include in the conformity assessment procedure a recommendation for indoor radon re-testing, such as every 3–10 years, depending on the RPA ranking [13]. The higher the rank of the RPA, the more often the repeated indoor tests should be carried out.

According to criterion (1), conformity assessment of a room with a norm through measurements covering all rooms with long-term occupancy allows the identification of a hazardous building as a whole and facilitates decisions regarding the need for mitigation activities. The same testing procedure within criterion (1) can be used to verify the effectiveness of mitigation using the short-term measurements.

Perhaps there are consistent patterns in the spatial behavior of radon across different rooms and/or floors in a building that could be statistically justified to enhance the applicability of criterion (1) for the entire building. However, all previously conducted [35,36,37,38] and planned studies (within the RadoNorm project) of spatial variations of radon within buildings still lack rigorous justification and clear outcomes from the perspective of the actual needs for indoor radon metrology and measurement standardization. Moreover, if the problem of conformity assessment of a room, as an elemental spatial and structural component of a building, remains not fully resolved, then the more comprehensive task of conformity assessment for an entire building solely based on patterns of spatial radon behavior evidently cannot have an effective solution.

## 3. Temporal Uncertainty of Indoor Radon

The investigation of temporal uncertainty, *U_V_*(*t*), considering the measurement duration and the influence of various factors on radon behavior over time, is a pertinent task in the metrological support of indoor radon measurements. From the description above, it is evident that the assessment of temporal uncertainty is based on a statistical analysis of deviations in indoor radon concentrations from annual average levels, measured in diverse conditions across a large number of buildings. This is a straightforward statistical approach to evaluating *U_V_*(*t*), yet it allows for the consideration of the impact of all anthropogenic and natural factors (including seasonal variations in indoor radon) on temporal uncertainty, if a representative sample of monitored buildings and rooms is provided.

An essential condition for ensuring data representativeness is the adequate selection of experimental buildings and rooms, for which the requirements are outlined in [27]. These requirements are further specified below:

(a)In each building, only one monitored room with an elevated activity concentration of radon (on average, at least 50–70 Bq m^−3^ that exceeds by at least 5 times the outdoor radon concentration) should be present. The conformity assessment of a room with a norm at low (close to outdoor) concentration is less demanding than with elevated indoor radon. Additionally, the pattern of radon temporal variations in buildings with low and elevated concentrations may differ due to the varying contribution of outdoor radon to the overall indoor radon balance.(b)The monitored room should be the one occupied for the longest duration (e.g., a bedroom or office space, occupied for at least 6 h a day).(c)Preference is given to both low-rise and high-rise dwellings with natural ventilation because there are significantly more buildings with natural ventilation than with forced ventilation.(d)The monitored room, along with the building itself, must operate in a regular (normal) mode throughout the entire monitoring period, excluding any special effects from natural (or mechanical) ventilation, as well as avoiding room or building repairs.

The algorithm for determining *U_V_*(*t*) is presented in the previous section. For more detailed information on the algorithm (including illustrations), please refer to [27]. In this section, we will discuss the original method of converting original YLCM data, which allows one to create powerful arrays of statistical data.

*C_i_*(*t*) is a set of transformed data arrays of the original YLCM in the same room, which differ in the integration interval *t*, for example, from 2 days to 11 months. It is important that each of the new arrays includes the same number of data (*i* = 1…*M*) as the original YLCM.

As an example, let us take the original YLCM data series with a registration (integration) interval of 3 h (*M =* 365 × 24/3 = 2920), which we transform into a time series with an integration interval of 2 days, obtaining an additional data array *C_i_*(*t* = 2 days). Figure 1 shows an example of a scheme for such a transformation through a “moving average” with a shift of 3 h.

Thus, a single original YLCM array allows for the generation of extensive arrays of additional *D_i_*(*t*) values. The total number of these, represented as *L*, is determined by the formula:(2)L=M⋅I
where *M* represents the number of data in the original YLCM array, and *I* represents the number of transformed data series (new arrays) with differing integration intervals.

The primary challenge in implementing the rational criterion is the lack of comprehensive knowledge about the temporal uncertainty of indoor radon *U_V_*(*t*) under various conditions such as climate, geology, architectural style, and other characteristics of different types of buildings and rooms. At present, our dataset is not sufficiently representative, covering only 6 and 12 YLCMs in Russia [32] and Israel [10], respectively. However, it includes more than 1.2 million *D_ij_*(*t*) values [27], which is approximately 25 times more than the combined data from other studies in the US [39] and Finland [40]. In this context, it is worth noting that, apart from these two studies, we have so far been unable to find additional useful data published by other researchers. For instance, a very recent study at the Harvard T.H. Chan School of Public Health [41], unfortunately, also lacks results that could be utilized to refine temporal uncertainty and advance the rational metrology of indoor radon measurements.

As shown in Figure 2, we have suggested indicative temporal uncertainty values [27], taking into account useful data from the US and Finland studies mentioned above. These temporal uncertainty values can be used for standardizing indoor radon measurements at an international level as part of a conservative approach, while the alternative data are absent.

The conservative approach implies a deliberate overestimation of uncertainty to ensure reliable decision making due to the insufficient representativeness (or power) of statistical data. For a more accurate (representative) assessment of indoor radon temporal uncertainty, it is necessary to conduct as many YLCMs as possible in various buildings located in different regions and countries. As the number of YLCMs increases under controlled conditions (covering the most significant influencing factors from a list of parameters related to climate, geology, and the characteristics of the room and building itself, including the location), the possibility of reducing the temporal uncertainty increases, using the pattern of influence of one or several factors at once if they show a strong correlation with *U_V_*(*t*). According to our estimates, conducting (again—under controlled conditions) YLCMs in 200–300 different buildings in Europe (or at least 20 YLCMs in each country) will allow us to obtain a statistically representative array of deviations *D_ij_*(*t*) for the verification and refinement of the conservative values of the temporal uncertainty *U_V_*(*t*) [10,16,27]. Such a study could easily be conducted within the framework of the already-mentioned RadoNorm project. Unfortunately, the coordinators of this project did not plan such an important activity.

In the articles [10,27], it is reasonably reported that unprofessional (inexpensive) continuous radon monitors can be successfully used for conducting YLCMs, and high detector sensitivity is not required, while the operation reliability of the devices is important. For example, the University of Galway (Ireland) recently conducted 348 YLCMs using “Airthings|Wave Plus” monitors (https://www.airthings.com/wave-plus, accessed on 12 January 2024). YLCMs were conducted during the COVID-19 quarantine period for 18 months, covering 87 buildings, which were spread across Ireland, including RPA. The results showed that about half of the buildings had elevated levels of radon. Preliminary analysis of the raw data, including the processing of several YLCMs for the calculation of *U_V_*(*t*), testified to the reliability of the huge array of raw data, which have not yet been processed. It is important to note that the cost of such a study in Ireland was less than 1% of the RadoNorm budget [42] or was significantly lower than the costs of data collection, for example, in study [21], which is mentioned above.

## 4. Instrumental Uncertainty

The instrumental uncertainty *U_D_* is a better-known parameter compared to the temporal uncertainty, as discussed above. However, the outputs of the earlier mentioned MetroRADON project [35], the ISO 11665-8 standard [26], and the latest release of US radon standards [28] are lacking universal, clear, and simple algorithms for determining *U_D_*, as well as adequate restrictive requirements for conformity assessment. In the context of metrological support for criterion (1), the calculation of *C*(*t*) and *U_D_* requires us to know the values of two basic metrological parameters, such as the sensitivity and background of the device (including those of the detector, as well as of the sampler, if any). These values are determined using the following algorithm:(3)$$C(t)=\frac{n_{g}/t-n_{0}/t_{0}}{\varepsilon }=\frac{r_{g}-r_{0}}{\varepsilon }\;$$
where *C*(*t*) is the calculated radon concentration obtained during the measurement period *t*, Bq m^−3^.

*n_g_* or *r_g_* is the number of counted pulses or the count rate (1/s) of the gross effect obtained during the measurement period *t*.

*n*_0_ or *r*_0_ is the number of counted pulses or the count rate (1/s) of the background effect obtained during the background measurement period *t*_0_, according to the instructions of the device manufacturer.

*ε* is the sensitivity of the device or calibration factor, expressed through the net count rate per Bq m^−3^ or 1/(s Bq m^−3^); this parameter can be determined using (3) if the reference value *C*(*t*) is known.

The uncertainty of the calculation of *u*(*C*) can be determined by applying the fundamental rules of ISO/IEC for the evaluation of measurement uncertainty [29] in relation to (3).
(4)$$u(C)=\sqrt{\frac{u^{2}(r_{g})+u^{2}(r_{0})}{\varepsilon^{2}}+u_{rel}^{2}(\varepsilon )\cdot {\left(\frac{r_{g}-r_{0}}{\varepsilon }\right)}^{2}}=C\cdot \sqrt{\frac{u^{2}(r_{g})+u^{2}(r_{0})}{(r_{g}-r_{0}{)}^{2}}+u_{rel}^{2}(\varepsilon )\;}$$
where *u*(*…*) and *u_rel_*(*…*) represent the uncertainties of the parameter in brackets, expressed in absolute and relative units, respectively. It should be noted that Equation (4) is consistent with examples in the ISO 11665 series and ISO 11929-4 [43,44,45], which regulate the measurement and calculation of activity (including radon concentration) using various methods.

If the distribution of counted pulses follows a normal law, which is valid when measuring ionizing radiation, then the following relationship can be used [29]:(5)$$u\left(r\right)=\frac{u\left(n\right)}{t}=\frac{\sqrt{n}}{t}=\sqrt{\frac{r}{t}\;},$$

Hence, also considering *U_D_* = *k · u(C)/C*, Equation (4) can be written as follows [45]:(6)$$U_{D}=k\cdot \sqrt{\frac{r_{g}/t+r_{0}/t_{0}}{(r_{g}-r_{0}{)}^{2}}+u_{rel}^{2}(\varepsilon )\;},$$
where *k* is the coverage factor equal to 2.

The first term under the root in Equation (6), expressed as a fraction, represents the random component of instrumental uncertainty, which depends on the duration of the measurements. The last term under the root in Equation (6) represents the systematic component of instrumental uncertainty related mainly to calibration, as well as other factors affecting only instrumental uncertainty, excluding the random component. If the detector background is absent or negligibly small, then *r*_0_ = 0 or *n*_0_ = 0.

It is important to note that Formulas (3) and (6), as well as the criterion itself (1), are universal and can be applied to different measurement methods and types of radon devices, including charcoal and electret methods, as well as solid-state nuclear track detector (SSNTD) and continuous radon monitor (CRM) methods. Such a universal metrological scheme for ensuring the uniformity of measurements and conformity assessment can be easily adapted even in relation to the CD/DVD method for retrospective measurements of the activity concentration of indoor radon [46]. Since CD/DVD and SSNTD methods are based on similar principles for measuring radon concentration, it is useful to show the adaptation of the metrological scheme for these methods. Here, it is not the count rate that is to be measured, but the total number of etched tracks or track density (with adjustment of the *ε*-value), that can be denoted by the same symbols *n_g_* and *n*_0_ (considering the background). In this case,
(7)$$C(t)=\frac{n_{g}-n_{0}}{\varepsilon \;t}\;,$$
and, also taking into account (5) and (6), we obtain
(8)$$U_{D}=k\cdot \sqrt{\frac{n_{g}+n_{0}}{(n_{g}-n_{0}{)}^{2}}+u_{rel}^{2}(\varepsilon )+u_{rel}^{2}(t)\;},$$
where the systematic component $u_{rel}^{2}\left(\varepsilon \right)\;$can cover a whole set of factors (humidity, air dust, ageing and fading effects, etc.), the influence of which leads to an increase in calibration uncertainty for long-term exposure conditions.

In terms of metrological challenges, the CD/DVD method holds a significant advantage by eliminating the temporal (key) uncertainty of indoor radon (*U_V_(t) =* 0) as its exposure duration consistently exceeds one year. At the same time, per Equation (8), an additional component of instrumental uncertainty arises from determining the exposure duration of CD/DVD discs, which may have been stored indoors for several years without requiring radon measurement. Laboratory analysis of the disks only takes a few days [46].

## 5. Requirements for Indoor Radon-Measuring Devices

The data analysis in Figure 2 shows that the most significant reduction in temporal uncertainty is observed in the measurement duration interval up to two days. Then, in the interval from 2 to 7 days, the reduction is less sharp, and in the measurement duration interval of more than 14 days, the rate of reduction of temporal uncertainty becomes very small. This important experimental observation explains the appropriateness of conducting short-term measurements in the interval from 2 (or better from 4) to 7 days. Measurements over several minutes or hours (so called “spot” measurements), which, in Figure 2, correspond to *t =* 0, are characterized by too-high (more than 200%) and uncontrollable temporal uncertainty, so they cannot be used, in principle, for conformity assessment, according to the world practice of indoor radon testing.

Since the duration of measurements should exceed two days (according to the US practice), the relevance of using radon devices with high sensitivity, which usually increases their cost, decreases. Indeed, *U_V_*(*t*) > 1.0 (100%) or *U_V_*(*t*) > 0.7 (70%), if the duration of measurements is no more than 2 or 7 days, respectively. Therefore, the reliability of the conformity assessment in the mode of short-term measurements will almost not decrease (or will not significantly decrease), even at *U_D_* = 0.4 (40%) within the framework of the rational criterion (1), because $\sqrt{1.0^{2}+0.4^{2}}\;$~ 1.0, and $\sqrt{0.7^{2}+0.4^{2}}\;$~ 0.8. Obviously, with an increase in the duration of measurements, the random component of uncertainty *U_D_* will decrease, according to (6), completely losing its role, for example, with a test duration of more than one or several weeks. In this case, *U_D_* will be determined only by the value of the systematic (calibration) component of uncertainty. In this regard, it is important to note that such common characteristic limits of devices or measurement methods, as the “detection limit” (or “minimum measurable activity”), as well as the “decision threshold” and “limits of the coverage interval”, according to ISO 11665 series [43,44] and ISO 11929-4 [45], completely lose their meaning, since, within the framework of the rational criterion for conformity assessment (1), there is no need to measure low radon activity concentration with high accuracy, especially in the mode of short-term measurements. In the mode of long-term measurements, the minimum measurable activity will tend to zero (or to the level of unavoidable activity concentration due to the influence of background or other interference) with an increase in the test duration if the systematic component of instrumental uncertainty is excluded. Therefore, in addition to (or even instead of) striving to reduce the minimum measurable activity, manufacturers of measuring equipment, especially CRM devices, are strongly recommended to display, as a result of measurement, not only the value of *C*(*t*), but also the current calculated value of *U_D_*, according to (3) and (6), finally introducing the fundamental concepts of ISO/IEC [29,30,31] into the practice of indoor radon measurements. The indication of these parameters is very convenient both when assessing conformity according to criterion (1), and when certifying or metrological-checking radon devices, especially during measurements.

For the purpose of metrological support, manufacturers of radon measurement devices are also strongly recommended to additionally display (for example, in the checking mode) the parameter *r* (or *n*) for determining key metrological parameters such as *r*_0_ (*n*_0_), *ε*, *u_rel_(ε)*, and also *U_D_*. These parameters are distinguishing features of radon measurement devices, irrespective of the detection principle and measurement procedure. In this context, to assess *U_D_*, its value should be calculated using formulas (3) and (6), setting, for instance, *C*(*t*) = 100 Bq m^−3^ at *t* = 1 and/or 24 h.

Another feature of criterion (1) is the possibility of rational management of the *U_D_* parameter [27], which was already discussed at the beginning of the section. For example, it is quite acceptable that the *U_D_* values can be several times higher for short-term measurements compared to long-term ones. This relaxation of metrological requirements for short-term tests is justified due to very high *U_V_*(*t*) values for short-term tests, which gradually decrease as the test duration increases (Figure 2), approaching the *U_D_* values at the level of 40, 30, 20, 15, and 10%. This important fact not only is addressed to radon device manufacturers and National Metrology Institutions, but also legalizes the participation of non-professionals in indoor radon measurement at the stage of short-term tests (screening). Indeed, metrological requirements related to *U_D_* control in relation to the most-demanded short-term measurements should be less stringent than for long-term measurements. In this regard, it is quite permissible to allow a professional user (inspector) to independently control the quality of their radon devices for short-term measurements, but at least one or several of them must be periodically checked at the National Metrology Institute. By the way, our experience shows that even non-professional indoor radon monitors demonstrate quite reliable operation for 3–5 or even more years thanks to good progress in microelectronic technology. In addition, the introduction of online technologies significantly improves the quality of indoor radon measurements. For example, even schoolchildren successfully cope with radon measurements in their homes under the control of the online system “RadonTest” [47].

Professional radon devices are usually distinguished by their high cost due to the high sensitivity of the detector (or several identical detectors are used inside one device). However, increasing the sensitivity (or accuracy) of the device does not contribute to improving the quality and metrological support of indoor radon measurements if the rational criterion for conformity assessment (1), based on the fundamental concepts of ISO/IEC [29,30,31], is not taken into account. In this regard, an important feature of the rational criterion is the fact that the use of any method or device for measuring indoor radon is allowed, even with the lowest sensitivity. In this case, a longer test duration may be required, depending on the level of the measured radon concentration, taking into account the ratio between temporal and instrumental uncertainties. Such an innovative feature in indoor radon measurements, also implying dynamic control of the main components of decision-making uncertainty, motivates the production of a wider range of radon devices, focusing on cheaper tools. This, obviously, will contribute to reducing the cost of indoor radon measurements, and the reliability of decision making will increase.

## 6. Conclusions

Metrological support is an important part of measurement standardization. However, both in the US and International (expressing only the European approach) standards for indoor radon measurements, clear mathematical algorithms (criteria) and QA/QC procedures within such fundamental ISO/IEC concepts as “measurement uncertainty” and “conformity assessment” are still lacking.The rational criterion for conformity assessment of a room with a norm has been improved. This criterion, within the ISO/IEC concept, takes into account the main uncertainties of measurements caused by temporal variations in indoor radon and instrumental error. This ensures the optimal choice of measurement duration, as well as the required reliability (not less than 95%) of decision making regarding compliance with the norm.The conditions for collecting initial data have been detailed, and the statistical algorithm for determining the temporal (key) uncertainty of indoor radon behavior has been improved, and its conservative values have been presented. Rational control of indoor radon can first be provided on the basis of a conservative approach, since to verify and refine the existing indicative values of uncertainty of temporal radon variations, additional research is needed in 200–300 buildings located in different countries.For the first time, a simple mathematical algorithm has been clearly presented for determining instrumental uncertainty in the dynamic mode (during measurements), and within the framework of a rational criterion, universal metrological requirements have been formulated for different methods and devices used for indoor radon measurements. These achievements allow for a better understanding of the need for the standardization of indoor radon measurements and a more effective implementation of QA/QC within the ISO/IEC concept.The materials in this article are proposed to be used as a conceptual basis during the ongoing (unfortunately, without any progress for several years) revision of international standards for indoor radon measurements (including QA/QC), which is being carried out by the ISO/TC85/SC2/WG17 “Radioactivity measurements” working group.

## Figures and Tables

**Figure 1 sensors-24-00504-f001:**
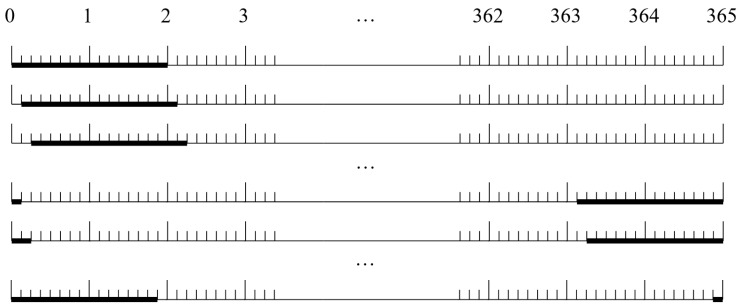
The scheme for converting the initial year-long continuous measurements (YLCMs) with a registration (integration) interval of 3 h into time series with the integration interval of 2 days (bold segments).

**Figure 2 sensors-24-00504-f002:**
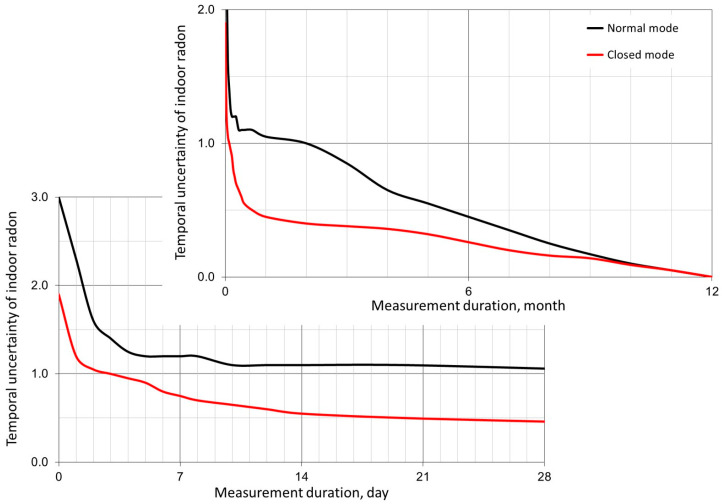
Indicative temporal uncertainty (relative) obtained for the normal and closed (windows and doors closed) modes of operation of a room, according to Table 4 from [27].

## Data Availability

Data are contained within the article.

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
