# Peer review of "Metrology for Indoor Radon Measurements and Requirements for Different Types of Devices"

_sensors, 2024, doi:10.3390/s24020504_

Round 1
Reviewer 1 Report
Comments and Suggestions for Authors
Dear authors, please see the attached docs:
- annotated manuscript
- review

Reviewer 2 Report
Comments and Suggestions for Authors
This work presents enhanced mathematical algorithms for determining both temporal and instrumental uncertainties. Also, unified metrological requirements are formulated for various methods and devices employed in indoor radon measurements. Authors express concern about the observed misalignment in priorities within the radon scientific community in terms of research and regulation of indoor radon. In this point, the authors thought that a more detailed discussion of these two main components of the combined uncertainty of annual average radon concentration would provide a better understanding of the metrological needs for the effective implementation of QA/QC into indoor radon measurements.
In general, the work is quite good and fits the journal's scope for the Section Physical Sensors in the Special Issue Detection and Measurement of Radioactive Noble Gases.
systematic way is used well and the authors present very good work.
Some minor revisions are required before acceptance
-Some English editing is required
Just examples
in the abstract line 17 (a specified) should be (specified)
line 110 (last) should be (the last)
line 123 (radon) should be (the radon)
line 126 (a specified) should be (specified)
- Lines 210-211 How do you determine this level (50-70 Bq m-3)? (In each building, only one monitored room with an elevated activity concentration of radon (on average, at least 50-70 Bq m-3 and exceeding the outdoor radon concentration at least 5 times) should be present).
-Figure 2. Indicative temporal uncertainty; could you present the concentration of radon as in Fig 2 for uncertainty?
Comments on the Quality of English Language-Some English editing is required
Just examples
in the abstract line 17 (a specified) should be (specified)
line 110 (last) should be (the last)
line 123 (radon) should be (the radon)
line 126 (a specified) should be (specified
Reviewer 3 Report
Comments and Suggestions for Authors
Dear Authors,
The manuscript "Metrology for Indoor Radon Measurements and Requirements for Different Types of Devices" focuses on standardizing indoor radon measurements. It highlights the absence of a globally harmonized measurement standard and the discrepancy between measurement protocols in the US and the European Union. The authors propose enhanced mathematical algorithms for determining temporal and instrumental uncertainties and formulate unified metrological requirements for various methods and devices used in indoor radon measurements.
The manuscript is generally well-written with a clear structure and understandable language. However, some points needs the authors consideration.
Detailed Comments and Suggestions:
1.Introduction
Comments: The introduction sets a clear context for the study but lacks depth in historical and global perspectives on radon measurement standards. It focuses more on the problem than on the evolution of the field.
Please involve bellow information as well:
- Expand the introduction to include a brief history of indoor radon measurement practices globally.
- Introduce a discussion on the challenges faced by researchers and practitioners in this field.
- Include a few more recent studies to show the current state of research in this area.
2.Literature Review and References
Comments: The references are relevant and current, but the literature review seems somewhat limited in scope.
- Please, broaden the literature review to include studies from different regions of the world.
- Also Compare and contrast various existing methodologies for radon measurement.
- Discuss any conflicting findings or theories in the literature to provide a comprehensive overview.
3.Research Design:
- The research design is appropriate but lacks detail on how the proposed methodologies might perform under varied real-world conditions.
- Discuss about how the proposed methods can be adapted or applied in different indoor environments, e.g. caves or mines or underground workplaces or places with good ventilation or poor ventilation.
- Mainly talk about limitation, discuss potential limitations or variables that could affect the accuracy of the measurements.
4.Methodology
- This part managed well and could address most of the concerns, but the connection between the theoretical models and practical application is not fully clear.
- It is necessary to provide a brief step-by-step examples of how the proposed algorithms can be implemented in practice.
- More explanation about any specialized equipment or conditions required for the methodology, is required.
- Additionally, please address potential challenges in implementing these methods and how they can be overcome.
5.Results
The results are informative and presented well.
- But it would be more beneficial to use more charts, graphs, and tables to visually represent the data.
- You should include case studies or examples to illustrate how the results apply in practical scenarios.
6.Conclusions
Well done, it could logically link from the results.
Overall Manuscript:
The manuscript is technically sound and contributes to the field.
The manuscript is recommended for publication after major revisions focusing on these aspects.
Some sections, particularly those involving complex methodologies and statistical analysis, are dense and could be broken down for better clarity.
Round 2
Reviewer 3 Report
Comments and Suggestions for Authors
No change in manuscript has been done and manuscript is same as original. My opinion is rejection.
good luck.
Author Response
The third reviewer incorrectly asserts that 'no change in manuscript has been done and manuscript is same as the original' (maintaining the original spelling). Contrary to this statement, changes have indeed been made. While the authors didn't accept many of the general comments from this reviewer, they provided detailed explanations about the reasons. Unfortunately, the reviewer did not engage with the authors' point-by-point responses to each of his/her comments.
Moreover, in the second review, the reviewer mechanically aligned all checkmarks in a single column, leading to a notably lower overall manuscript rating (compared to the first review, where the marks were placed in a checkerboard pattern). This is despite the reviewer's claim that no changes were made.
Regrettably, these signs suggest a lack of expertise in reviewing research papers, both in this specific field and likely in others. In addition, a conflict of interest may be at play, as we critique well-known European projects like MetroRADON and RadoNorm that, unfortunately, do not serve the needs of metrology and standardization for indoor radon measurements.